# Therapeutic monitoring of adalimumab at non-trough levels in patients with inflammatory bowel disease

**Masaichi Kato[1], Ken Sugimoto[2]\*, Kentaro Ikeya[1], Ryosuke Takano[1], Ai Matsuura[1], Takahiro Miyazu[2], Natsuki Ishida[2], Satoshi Tamura[2], Shinya Tani[2], Mihoko Yamade[2], Yasushi Hamaya[2], Moriya Iwaizumi[3], Satoshi Osawa[4], Takahisa Furuta[5], Hiroyuki Hanai[1]**

1 Center for Gastroenterology and Inflammatory Bowel Disease Research, Hamamatsu South Hospital, Hamamatsu, Japan, 2 First Department of Medicine, Hamamatsu University School of Medicine, Hamamatsu, Japan, 3 Department of Laboratory Medicine, Hamamatsu University School of Medicine, Hamamatsu, Japan, 4 Department of Endoscopic and Photodynamic Medicine, Hamamatsu University School of Medicine, Hamamatsu, Japan, 5 Center for Clinical Research, Hamamatsu University School of Medicine, Hamamatsu, Japan

\* sugimken@hama-med.ac.jp

**Data Availability Statement:** All relevant data are within the paper and its Supporting Information files. Additional data related to this study may be requested from the authors.

## Abstract

Adalimumab (ADA) trough level and anti-ADA antibody (AAA) positivity influence mucosal healing and loss of response in patients with inflammatory bowel disease (IBD). In this study, we clarified the correlation between ADA monitoring, including non-trough level, and real-world IBD clinical outcomes. This retrospective, observational, single-center study involved patients with ulcerative colitis (19) and Crohn's disease (33) treated with ADA from January 2007 to August 2018. Serum ADA and AAA levels were measured 4–14 days after ADA administration. The AAA positivity rate was 23.1% (12/52). ADA continuity was higher in AAA-negative patients than in AAA-positive patients ($P = 0.223$). Receiver operating characteristic (ROC) analysis revealed that a serum AAA cut-off of 9.2 µg/mL was associated with ADA continuity. The ADA level was significantly higher in the endoscopic remission group than in the non-remission group ($P = 0.02$). Based on the ROC curve analysis results of serum ADA level and endoscopic remission, the cut-off value of the serum ADA level was set to 11.1 µg/mL. Under the combined use of ADA with immunomodulators and AAA positivity, ADA continuity was significantly higher when the serum AAA level at 4–14 days after ADA administration was ≥9.2 µg/mL. Furthermore, endoscopic remission can be expected with a serum ADA level of ≥11.1 µg/mL. Overall, to predict clinical outcomes, it would be useful to measure the blood level of ADA regardless of the timing of the trough.

## Introduction

Although the etiology of refractory inflammatory lesions in the intestinal tract of patients with inflammatory bowel diseases (IBDs), such as ulcerative colitis (UC) and Crohn's disease (CD), is unknown, inflammation caused by immune system dysregulation in the intestinal mucosa

**Funding:** No external funding was received for this study.

**Competing interests:** The authors have declared that no competing interests exist.

has been postulated to be the underlying cause of IBD pathogenesis [1]. The mainstay treatment for IBD is the suppression of immunocompetent cells by the systemic administration of steroids [2]. However, during recent years, biologics targeting cytokines produced by immunocompetent cells have been increasingly used in IBD treatment [3]. Among the biologics, anti-TNF-α agents are widely used for IBD. In Japan, infliximab (IFX) and adalimumab (ADA) are approved for CD treatment and golimumab is approved for UC treatment.

Biologics, such as IFX and ADA, not only induce clinical remission in patients with IBD but also cause mucosal healing [4–6]. However, loss of response (LOR) is a major concern in IBD management, as it occurs within 1 year in an estimated 13% of patients treated with IFX [7] and 20% of patients treated with ADA [8]. Although the mechanism of LOR is not clear, immunogenicity against anti-TNF-α inhibitors, manifested as a decrease in the serum level (trough level) of anti-TNF-α owing to the production of anti-drug antibodies or undetectability of anti-TNF-α, is involved in the LOR5.

The importance of maintaining a high trough level of anti-TNF-α agents has been shown in previous studies, which demonstrated that a low IFX trough level makes it difficult to achieve mucosal healing [4, 5, 9]. As reported, the risk of LOR in patients with antibodies against IFX (ATI) is three times that of patients without ATI [10]; it is important to suppress the production of anti-drug antibodies. Both trough level and anti-ADA antibody (AAA) have been associated with mucosal healing and LOR [11]. During recent years, therapeutic drug monitoring (TDM), a strategy to optimize treatment by measuring the serum trough level and anti-drug antibody level, has been recommended to maximize the effects of these biological agents [12]. There are two types of TDM: reactive TDM that promotes decision making for patients who show treatment failure and proactive TDM that optimizes treatment and potentially avoids flare and LOR in patients who have showed some therapeutic response [13]. However, the treat-to-target strategy, which has been advocated in recent years, emphasizes the importance of proactive TDM in the treatment of patients with IBD in clinical remission [14]. A review by Mitrev et al. [12] recommends steady-state trough levels between 3 and 8 μg/mL for IFX and between 5 and 12 μg/mL for ADA. The upper limit of the therapeutic window of ADA trough level to achieve clinical remission in patients with IBD has not been defined.

Another problem in real-world clinical practice of TDM is that the regular visit dates of patients do not always coincide with the trough measurement date, that is, 14 days after ADA administration. It is not always possible to schedule another consultation visit only for the measurement of the trough level or to change the time of injection to measure the trough level during the consultation visit. Interestingly, Ward et al. [15] reported that the trough level could be estimated from serum ADA level from days 3–9 after ADA injection, suggesting the feasibility of TDM even without trough level measurement of ADA. In addition, the authors reported that 8 weeks after the initiation of ADA therapy, the average fluctuation in the ADA level during the 2 weeks between successive administrations of ADA was approximately 3 μg/mL [16]. Therefore, the purpose of our study was to analyze the effects of serum ADA levels, including its non-trough levels, and AAA presence on the subsequent treatment outcomes in patients with IBD receiving ADA. We also aimed to clarify the correlation between TDM for ADA and clinical outcomes in a real-world setting.

## Materials and methods

### Patients

Fifty-two patients with IBD (UC or CD) were enrolled in this study and treated with ADA at Hamamatsu South Hospital, Hamamatsu, Japan, from January 2007 to August 2018. Serum drug (ADA) and AAA levels were measured during TDM. We screened medical records in

October 2018 and identified patients to be included in the study in November 2018. All patients provided informed consent for the use of medical records before enrolling in this study from December 2018 to January 2019. Patients with Behcet's disease, indeterminate colitis, and unclassified IBD were excluded from the study. ADA was prescribed at standard doses of 160 mg at week 0, 80 mg at week 2, and 40 mg subsequently every other week for maintenance; the drug was administered subcutaneously. The use of adalimumab described in our study was implemented as standard-of-care at our hospital. In addition, assignment of patients to adalimumab was made at the discretion of the treating physician. The observation period was from the start of ADA administration to ADA discontinuation, and observations were continued until August 2018 in patients who were still on ADA in August 2018. We measured the anti-drug antibody and drug levels at the visit dates of patients; we excluded patients whose antibody and drug levels were measured within 3 days of ADA administration. In all cases, the anti-drug antibody was measured at the same timing as the drug concentration.

## Study design

This was a retrospective, observational single-center study. The primary endpoint of this study was the comparison of treatment continuity between (i) the AAA-positive and -negative groups and (ii) the high- and low-serum ADA groups. The secondary endpoints were the (i) comparison of endoscopic remission rate between patients with high- and low-serum ADA levels and (ii) extraction of factors that may influence serum ADA level. Treatment continuity implied that neither the dose nor the dosing interval of ADA was changed during the observation period. Accordingly, patients who discontinued ADA therapy or received treatment enhancements, such as double-dose administration and shortened ADA administration period, did not fit this description. The discontinuation of ADA because of transferring patients to another hospital, the preference of patients, or discontinuation because of adverse events was not considered "censored." Endoscopic remission was defined as a Mayo endoscopic score of ≤1 in patients with UC and a simple endoscopic score for CD of ≤3 in patients with CD. Endoscopic evaluation was conducted only for patients who underwent endoscopy after introducing ADA.

## Laboratory methods

We measured the serum level of ADA and AAA (free and bound) using enzyme-linked immunosorbent assay (ELISA) (IDKmonitor® Adalimumab drug level ELISA and IDKmonitor® Adalimumab total ADA ELISA; Immundiagnostik AG, Germany).

## Statistical analysis

The Kaplan–Meier estimator was used to generate survival curves for the AAA-positive and -negative groups. The log-rank test was used to compare persistence rates (treatment continuity) with ADA (the drug) between strata. A receiver operating characteristic (ROC) curve analysis was performed for serum drug levels to identify the cut-off value associated with treatment continuity. The two groups of patients were divided into a low-drug level group below the cut-off value and a high-drug level group above the cut-off value. We used the Kaplan–Meier estimator to generate survival curves for the high-drug level and low-drug level groups and compared persistence rates with ADA between strata using the log-rank test. Next, the ROC curve analysis was performed to determine the cut-off value of serum drug level for endoscopic remission. We divided the patients into two groups, the low-drug level group below the cut-off value and the high-drug level group above the cut-off value, and compared the endoscopic remission rate between the groups using the $\chi^2$ test. A univariate analysis was

performed to identify factors affecting the serum ADA level via Mann–Whitney *U* test, Pearson's correlation coefficient test, and Spearman's correlation coefficient using the log-rank test. Factors included in the model were sex, age, body weight, IBD type (UC or CD), disease duration, the time from diagnosis to first ADA administration, the period from the start of ADA therapy to TDM implementation, IFX treatment history, and concomitant use of immunomodulators. These data were obtained when TDM was performed on the aforementioned factors. All statistical analyses were performed using EZR (Saitama Medical Center, Jichi Medical University, Saitama, Japan), which is a graphical user interface for R (The R Foundation for Statistical Computing, Vienna, Austria). Results with $P < 0.05$ were considered statistically significant.

### Ethical declarations

The study protocol was reviewed and approved by the Committee on the Ethics of Clinical Trials Involving Human Subjects at Hamamatsu South Hospital. In addition, the trial adhered to the principles of Good Clinical Practice and the ethical standards stipulated in the 1964 Declaration of Helsinki and its subsequent amendments. From December 2018 to January 2019, we provided patients with an explanation and consent for this study, and written informed consent was obtained from all patients after explaining the purpose of the study and the use of medical records and nature of the procedures to them. We were able to obtain consent from all patients for the use of medical records. The study also used an opt-out consent mechanism, and the Committee on the Ethics of Clinical Trials Involving Human Subjects at Hamamatsu South Hospital approved this method of consent. If subjects were under 18, informed consent was obtained from a parent and/or legal guardian.

## Results

### Patient characteristics

Nineteen patients with UC and 33 patients with CD were enrolled in this study (Table 1). Twenty-six of the 52 patients (50.0%) were previously treated with IFX. Twenty-seven of the

**Table 1. Baseline characteristics of patients.**

|  | N = 52 |
|---|---|
| UC:CD, n | 19:33 |
| Male: female, n | 38:14 |
| Age, median (IQR) | 39 (17–70) |
| Disease duration, year, median (IQR) | 10.5 (1–44) |
| Time from diagnosis to the first ADA administration, year, median (IQR) | 8 (0–41) |
| Period from ADA initiation to TDM implementation, week, median (IQR) | 80 (2–427) |
| Body weight, kg, median (IQR) | 55.5 (37–82) |
| IFX treatment history, n (%) | 26/52 (50%-0%) |
| Combination treatment |  |
| 5-ASA, n (%) | 50/52 (96.2%) |
| Immunomodulators (AZA, 6-MP, n [%]) | 27/52 (52.0%) |
| Steroid (systemic), n (%) | 3/52 (5.8%) |

UC, ulcerative colitis; CD, Crohn's disease; IQR, interquartile range; ADA, adalimumab; TDM, therapeutic drug monitoring; IFX, infliximab; IQR, interquartile range; 5-ASA, 5-aminosalicylic acid; AZA, azathioprine; 6-MP, 6-mercaptopurine.

52 patients (52.0%) received immunomodulators in combination with ADA, and three of the 52 patients (5.8%) were treated with corticosteroids. According to the Montreal classification, we observed: (i) for UC, E1: 0 cases, E2: 4 cases (21.1%), E3: 15 cases (78.9%); (ii) for CD, L1: 6 cases (18.2%), L2: 4 cases (12.1%), L3: 23 cases (69.7%), B1: 14 cases (42.4%), B2: 12 cases (36.4%), and B3: 7 cases (21.2%). In addition, 17 patients (51.5%) with CD had previously undergone enterectomy. In this study, we measured the serum concentrations of ADA and AAA at any time, and the median time of TDM was 11 days (4–14) after ADA administration.

## AAA and treatment continuity

Here, AAA positivity was observed in 12 of the 52 patients (23.1%): 6 of the 19 (31.6%) patients with UC and 6 of the 33 (18.2%) patients with CD were positive for AAA. We found that treatment continuity was slightly lower in the AAA-positive group than in the AAA-negative group ($P$ = 0.223 by log-rank test; Fig 1). Next, we compared the endoscopic remission rate between the AAA-positive and AAA-negative groups, and no significant difference was observed between the groups ($P$ = 0.46, data not shown).

## ADA level and treatment continuity

Based on the results of the ROC curve analysis of serum ADA level performed to identify the cut-off value associated with treatment continuity, we set the cut-off value of the serum ADA level to 9.2 μg/mL (area under the curve [AUC] = 0.767, 95% confidence interval [CI] = 0.636–0.899, positive predictive value [PPV] = 72%, negative predictive value [NPV] = 74%, sensitivity = 0.600, specificity = 0.852, and accuracy = 73%; Fig 2A). Patients with serum ADA level of <9.2 μg/mL were assigned to the low-drug level group and those with ≥9.2 μg/mL to the

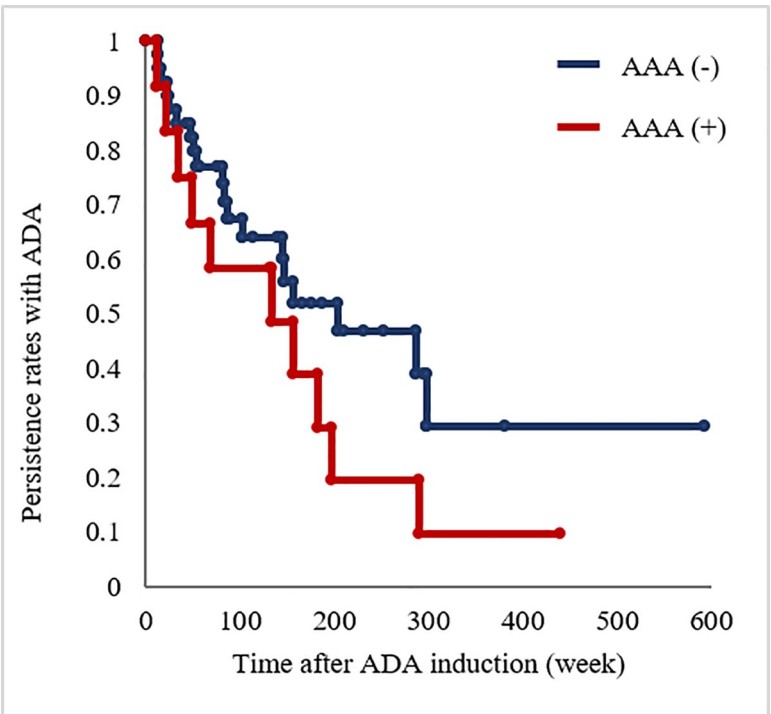

**Fig 1. Kaplan–Meier curves of time to ADA discontinuation in patients in the AAA-negative group vs. those in the AAA-positive group ($P$ = 0.023 by log-rank test).** ADA, adalimumab; AAA, anti-adalimumab antibody.

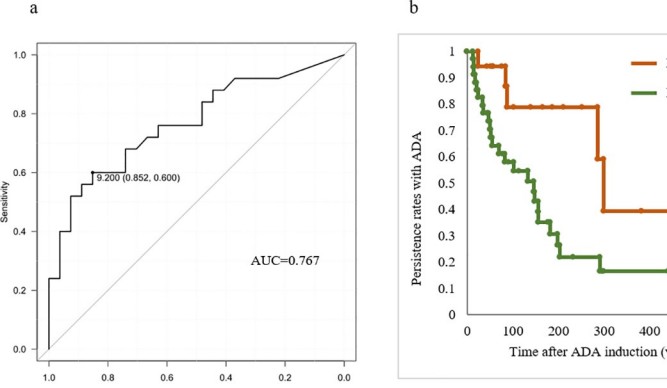

**Fig 2.** a. Receiver operating characteristic analysis for ADA level stratifying patients with and without ADA discontinuation. AUC, area under the curve. b. Kaplan–Meier curves of time to ADA discontinuation in patients in the high-drug level (ADA level ≥ 9.2 μg/mL) group vs. low-drug level (ADA level < 9.2 μg/mL) group ($P$ < 0.01 by log-rank test). ADA, adalimumab; AUC, area under the curve.

high-drug level group. Treatment continuity was significantly higher in the high-drug level group than in the low-drug level group ($P$ < 0.01, log-rank test; Fig 2B). We also analyzed cases with UC or CD alone. In the CD cases, the ADA continuation rate was significantly higher in the high drug level group than in the low drug level group (p = 0.003) (S1A Fig), but for the UC cases, there was no significant difference between the two groups (p = 0.190) (S1B Fig).

## ADA level and endoscopic remission

We analyzed the remission rate in patients who underwent endoscopy (44/52) after ADA administration. The median endoscopic severity of those 44 cases was MES 2 for UC (IQR 0–3) and SES-CD 9 for CD (IQR 0–25). Endoscopic remission was observed in 15 of the 44 patients (34.1%), and the serum ADA level was significantly higher in the endoscopic remission group than in the non-remission group (12.4 vs. 6.4 μg/mL, $P$ = 0.02; Fig 3A). An ROC curve analysis was performed to determine the optimal serum ADA level for achieving

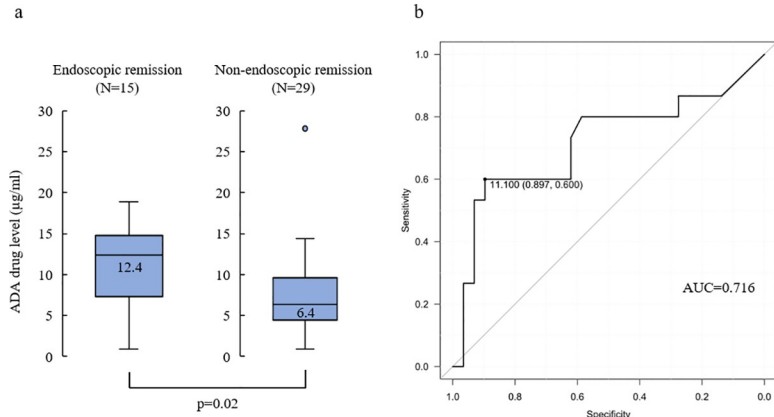

**Fig 3.** a. Distribution of ADA levels in patients with and without endoscopic remission. Box plots (5%–95%) show the median (solid line within the box), interquartile range (upper and lower box boundaries), and standard deviation (whiskers). There was a significant difference in the ADA level between patients with and without endoscopic remission ($P$ = 0.02 by Mann–Whitney $U$ test). b. Receiver operating characteristic curve analysis for ADA level stratifying patients with and without endoscopic remission. AUC, area under the curve; ADA, adalimumab.

endoscopic remission, and a serum ADA level of 11.1 μg/mL was identified as the cut-off value (AUC = 0.716, 95% CI 0.533–0.900, PPV = 75%, NPV = 81%, sensitivity = 0.600, specificity = 0.897, accuracy = 80%; Fig 3B). We then compared the endoscopic remission rate between the low-drug level group with a serum ADA level of <11.1 μg/mL and the high-drug level group with a serum ADA level of ≥11.1 μg/mL. A significantly higher number of patients achieved endoscopic remission in the high-drug level group (32/44, 72.7%) than in the low-drug level group (12/44, 27.3%; $P$ <0.01).

## Relationship between patient- and disease-related factors and drug levels

We performed a univariate analysis to identify the factors that may influence the serum ADA level (Table 2). None of the factors was found to be associated with the ADA level. We also compared treatment continuity with and without immunomodulators (combination therapy vs. monotherapy) but found no significant difference between the groups ($P$ = 0.40; Fig 4).

## Discussion

In this study, we showed that serum ADA levels, including non-trough values, but excluding values obtained within 3 days of ADA administration, reflect the subsequent clinical course and endoscopic outcome. Our study demonstrated that ADA continuity could be expected if the non-trough serum ADA level (except within 3 days of ADA administration) is ≥9.2 μg/mL and that endoscopic remission can be expected if the level is ≥11.1 μg/mL. A previous study on the trough level of ADA during maintenance therapy reported that 5–5.8 and 4.9–7.1 μg/mL were the lower limits of the therapeutic range when clinical remission and mucosal healing were the desired targets, respectively; both limits are approximately 4 μg/mL lower than our non-trough level. As these are the lower limits of the therapeutic range and the variation in the levels over 2 weeks between successive ADA administrations is approximately 3 μg/mL [16], our results are valid.

In other studies, AAA positivity was estimated to be 12.7–30.9% [9, 11, 15, 17], which is consistent with our results in this study (23.1%). Nakase et al. [18] reported that AAA and clinical remission were significantly correlated in patients with CD receiving ADA. In contrast, Robin et al. [17] showed that AAA positivity was higher in patients with clinically active disease than in patients with clinical remission (13.5 vs. 9.8 ng/mL, respectively). In addition,

**Table 2. Factors influencing blood ADA level.**

|  | *P*-value (univariate analysis) |
|---|---|
| Sex | 0.567[†] |
| Age | 0.860[‡] |
| Body weight | 0.570[‡] |
| UC or CD | 0.430[†] |
| Disease duration (years) | 0.585[§] |
| Period from diagnosis to ADA start (years) | 0.386[§] |
| Period from ADA start to TDM implementation (weeks) | 0.419[§] |
| IFX treatment history | 0.714[†] |
| Combination of immunomodulators | 0.252[†] |

UC, ulcerative colitis; CD, Crohn's disease; ADA, adalimumab; TDM. therapeutic drug monitoring; IFX, infliximab.

[†]Mann–Whitney's U test

[‡]Pearson's correlation coefficient test

[§]Spearman's correlation coefficient by rank test.

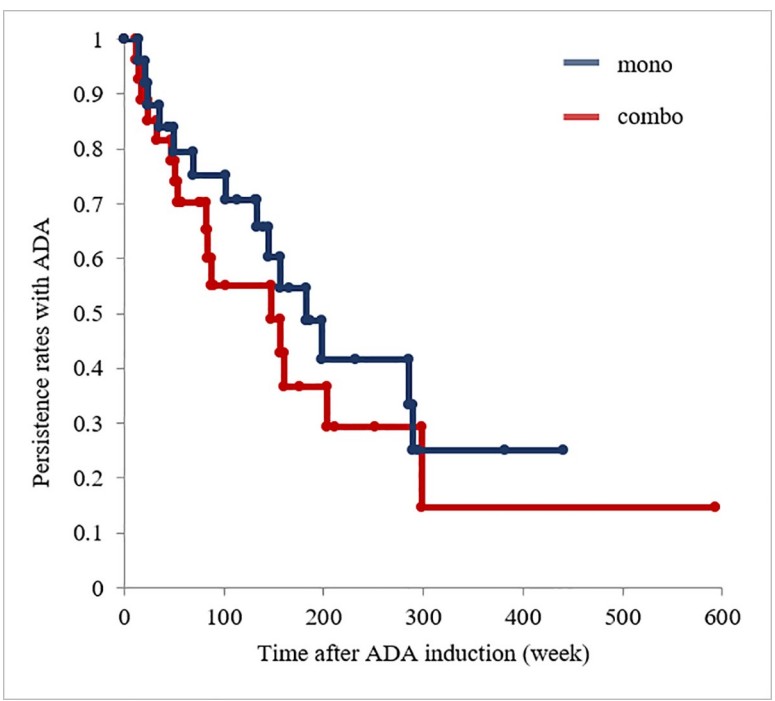

**Fig 4. Kaplan–Meier curves of time to ADA discontinuation in patients in the ADA monotherapy group vs. those in the ADA plus immunomodulator combination therapy group ($P = 0.40$ by log-rank test).** Mono, monotherapy; combo, combination therapy; ADA, adalimumab.

AAA positivity was lower in patients who achieved mucosal healing than in those who did not (6.5 vs. 14.1 ng/mL; $P = 0.06$). Our study also showed that AAA did not correlate with treatment continuity or endoscopic outcomes; hence, the importance of measuring the AAA level in clinical practice should be investigated further.

Reports on the benefits of the combined use of immunomodulators with ADA are controversial [19–24]. In this study, although there was no association between the concomitant use of immunomodulators with ADA and only ADA ($P = 0.252$), AAA positivity was significantly lower in the immunomodulator combination group than in the immunomodulator non-combination group (1/27 [3.7%] vs. 11/25 [44%], $P < 0.05$, Mann–Whitney $U$ test). However, there was no significant difference in treatment continuity with or without immunomodulators ($P = 0.40$). The use of immunomodulators suppressed AAA production, but it did not improve any clinical outcomes in our study. In the DIAMOND trial, a combination therapy of ADA and immunomodulator was more effective in mucosal improvement than ADA monotherapy.22 On the contrary, in our study, there was no significant difference in the rate of mucosal healing between the ADA monotherapy group and the ADA and IM combination therapy group ($p = 0.34$, $\chi 2$ test). However, a recent study reported that treatment continuity was higher with azathioprine when a second anti-TNF-α agent was used in cases where the use of the first anti-TNF-α agent resulted in immune-mediated LOR [25]. Further research is needed to determine whether it is necessary to use an immunomodulator with ADA in real-world clinical settings. In our study, because we found no other factors related to ADA level, we concluded that ADA is clinically effective if the serum ADA level is high, regardless of sex, weight, disease duration, and IFX treatment history. Therefore, to predict the therapeutic efficacy of ADA, the actual measurement of ADA level is important, and not the timing of blood collection to specifically measure the trough level (unless collected within 3 days of ADA administration).

A limitation of this study is that it was performed retrospectively and lacked records of clinical evaluations, such as CAI and CDAI. In general, ADA continuity is associated with the improvement of clinical symptoms. However, we were unable to assess the correlation between TDM levels and clinical symptoms. Another limitation is that the measurement time of ADA concentrations differed for each patient. There are a few studies on the pharmacokinetics of ADA in patients with IBD, but information about the changes in drug level after administration is limited. A *post-hoc* study on the pharmacokinetics of 65 patients with CD receiving ADA reported that the half-life of ADA was 22 days in AAA-negative patients [26]. A prospective observational study in which 19 patients with CD treated with ADA and underwent TDM at multiple timepoints indicated that the change in the serum ADA level over 9 days after administration was negligible [15]. Another prospective study on the pharmacokinetics of seven patients with CD undergoing remission maintenance therapy with ADA reported a minimal difference between the peak and trough blood ADA levels during ADA therapy [27]. From these studies, we inferred that the difference in the timing of TDM administration would have a negligible effect on the serum ADA level in patients repeatedly receiving ADA. This is because the ADA level in the blood increases and disappears relatively slowly because of the subcutaneous formulation, and the administration interval is 2 weeks, which is shorter than that for other drugs.

## Conclusions

In this study, we evaluated non-trough drug levels, which may be useful in predicting the outcomes of ADA therapy, unlike that with other biologics. Although this was a retrospective study based on actual clinical data, our study had a long observation period: median of 102 weeks and maximum of 593 weeks. We could not find any longer-term observational study that examined the relationship between the ADA level and its therapeutic effect.

In patients with IBD receiving ADA, a higher serum ADA level was associated with higher treatment continuity and endoscopic remission rates; however, AAA positivity did not affect ADA continuity. None of the factors we examined (such as the history of IFX administration and concomitant use of immunomodulators) was associated with the serum ADA level. Therefore, to predict clinical outcomes, it would be useful to measure the blood level of ADA regardless of the timing of the trough.

## Supporting information

**S1 Fig.** Kaplan-Meier curve of the time to ADA discontinuation in (A) UC and (B) CD patients within the high drug level group (ADA drug level ≥ 9.2 μg/ml) vs. the low drug level group (ADA drug level < 9.2 μg/ml; p<0.01 by Log Rank). In patients with UC, there was no significant difference in the ADA continuation rate between the high drug level group and the low drug level group (p = 0.19), but there was a significant difference in patients with CD (p<0.01).
(PPTX)

## Author Contributions

**Conceptualization:** Masaichi Kato, Ken Sugimoto, Satoshi Osawa, Hiroyuki Hanai.

**Data curation:** Masaichi Kato, Ken Sugimoto, Kentaro Ikeya, Moriya Iwaizumi, Satoshi Osawa, Hiroyuki Hanai.

**Formal analysis:** Masaichi Kato, Ken Sugimoto, Yasushi Hamaya, Moriya Iwaizumi, Hiroyuki Hanai.

**Funding acquisition:** Ken Sugimoto, Ryosuke Takano, Yasushi Hamaya.

**Investigation:** Masaichi Kato, Ken Sugimoto, Ryosuke Takano, Ai Matsuura, Takahiro Miyazu, Natsuki Ishida, Satoshi Tamura, Shinya Tani, Mihoko Yamade, Yasushi Hamaya, Moriya Iwaizumi, Hiroyuki Hanai.

**Methodology:** Ken Sugimoto, Ryosuke Takano, Ai Matsuura, Satoshi Osawa, Hiroyuki Hanai.

**Project administration:** Ken Sugimoto, Kentaro Ikeya, Ryosuke Takano, Ai Matsuura, Hiroyuki Hanai.

**Resources:** Ken Sugimoto, Kentaro Ikeya, Ryosuke Takano, Ai Matsuura, Takahisa Furuta, Hiroyuki Hanai.

**Software:** Kentaro Ikeya, Ryosuke Takano, Ai Matsuura, Yasushi Hamaya, Moriya Iwaizumi, Takahisa Furuta.

**Supervision:** Masaichi Kato, Ken Sugimoto, Kentaro Ikeya, Ryosuke Takano, Ai Matsuura, Shinya Tani, Yasushi Hamaya, Moriya Iwaizumi, Takahisa Furuta, Hiroyuki Hanai.

**Validation:** Kentaro Ikeya, Ai Matsuura, Takahiro Miyazu, Natsuki Ishida, Satoshi Tamura, Shinya Tani, Mihoko Yamade, Moriya Iwaizumi, Satoshi Osawa, Takahisa Furuta.

**Visualization:** Kentaro Ikeya, Ai Matsuura, Moriya Iwaizumi, Takahisa Furuta.

**Writing – original draft:** Masaichi Kato, Ken Sugimoto.

**Writing – review & editing:** Masaichi Kato, Ken Sugimoto, Satoshi Osawa, Hiroyuki Hanai.

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
