## [Decision Letter · Decision Letter 0]

24 May 2021

PONE-D-21-13113

Therapeutic monitoring of adalimumab at non-trough levels in patients with inflammatory bowel disease

PLOS ONE

Dear Professor Ken Sugimoto:

Thank you for submitting your manuscript to PLOS ONE. After careful consideration, we feel that it has merit but does not fully meet PLOS ONE’s publication criteria as it currently stands. Therefore, we invite you to submit a revised version of the manuscript that addresses the points raised during the review process.

Your manuscript was assessed by two expert reviewers in this field, and substantive issues have been raised as listed below: Regarding the criticisms from both reviewers, you must fully address the issues in the revised results and discussion sections. In particular, a separated analysis between UC and CD (e.g., UC alone, CD alone) seems to be critical, as mentioned by Reviewer 1.  In addition, you need to carefully mention the severity of UC as well as CD in the revised manuscript, as suggested by Reviewer 2.

We look forward to receiving your revised manuscript.

Kind regards,

Emiko Mizoguchi, M.D., Ph.D.

Academic Editor

PLOS ONE

Journal Requirements:

In the ethics statement in the manuscript and in the online submission form, please provide additional information about the patient records/samples used in your retrospective study, including: a) whether all data were fully anonymized before you accessed them; b) the date range (month and year) during which patients' medical records/samples were accessed; c) the source of the medical records/samples analyzed in this work (e.g. hospital, institution or medical center name).

3. We note that you obtained consents from participants to take part in your retrospective study. In the Ethics Statement on the online submission form and the manuscript Methods , please clarify the context in which consent was obtained, and specify whether patients provided:

                1) Consent to use their medical records/samples in research

                2) Consent to undergo the procedure

                3) Consent to take part in the study reported in this manuscript.

If the ethics committee waived the need for additional informed consent, please state this.

Thank you for your attention to these requests."

4. Please include the following information regarding your study design in your Methods:

 1) Whether the use of adalimumab described in your study was implemented as standard-of-care at your hospital.

2) Whether assignment of patients to adalimumab was made at the discretion of the treating physician or for the purposes of research."

5. Please note that PLOS does not permit references to 'data not shown.' Authors should provide the relevant data within the manuscript, the Supporting Information files, or in a public repository. If the data are not a core part of the research study being presented, we ask that authors remove any references to these data.

6. Thank you for stating the following financial disclosure:

7. We note that you have indicated that data from this study are available upon request. PLOS only allows data to be available upon request if there are legal or ethical restrictions on sharing data publicly. For information on unacceptable data access restrictions, please see http://journals.plos.org/plosone/s/data-availability#loc-unacceptable-data-access-restrictions.

Reviewers' comments:

Reviewer's Responses to Questions

**Comments to the Author**

1. Is the manuscript technically sound, and do the data support the conclusions?

Reviewer #1: Yes

Reviewer #2: Yes

2. Has the statistical analysis been performed appropriately and rigorously? 

Reviewer #1: Yes

Reviewer #2: Yes

3. Have the authors made all data underlying the findings in their manuscript fully available?

Reviewer #1: Yes

Reviewer #2: Yes

4. Is the manuscript presented in an intelligible fashion and written in standard English?

Reviewer #1: Yes

Reviewer #2: Yes

5. Review Comments to the Author

Reviewer #1: Although this is a retrospective, observational study, it reveals the interesting results that non-trough ADA levels may predict the outcomes of IBD patients. The research is presented in an intelligible fashion. Several points should be clarified.

All through the study: This type of studies has been usually conducted for a separated disease, such as UC alone, or CD alone, not for combined IBD like this study. Please describe the advantages and disadvantages of this method.

Line 275: I would like to know the reasons why AAA did not correlate with treatment continuity in this study, which is quite different from most prior reports. Due to AAA levels? combined evaluation of UC and CD? Please describe.

Reviewer #2: Kato et al. examined the correlation between anti-ADA antibody level and ADA continuity or endoscopic remission in UC and CD patients. Moreover, the authors examined the correlation between ADA level and ADA continuity or endoscopic remission. They found that a higher serum ADA level, including non-trough level, was associated with both higher ADA continuity and endoscopic remission rate, but level of AAA did not. Thus, they concluded that ADA level, including trough level and non-trough level, is useful for predicting clinical outcome of UC and CD patients treated with ADA.

There are some concerns about this study.

1. The authors evaluated the ADA level, including trough and non-trough. In each result, they should show the timing of measuring the ADA concentration (average, mean, etc.).

2. In Diamond study, combination of ADA and immunomodulator more effective on mucosal improvement that ADA monotherapy. They should show the endoscopic remission rate or endoscopic improvement rate in ADA monotherapy therapy and in ADA and IM combination therapy.

3. In general, the ADA continuity is associated with the improvement of clinical symptoms. Therefore, the ADA level may be associated with the improvement of clinical symptoms. They should show the correlation between the ADA level and the improvement of clinical symptoms.

Moreover, they should show the correlation between the positive or negative of AAA and the improvement of clinical symptoms.

4. The timing of evaluation of AAA level is missing.

5. The severity of UC and CD patients in this study is missing.

6. PLOS authors have the option to publish the peer review history of their article (what does this mean?). If published, this will include your full peer review and any attached files.

Reviewer #1: No

Reviewer #2: No

---

## [Author Response · Author response to Decision Letter 0]

18 Jun 2021

June 12, 2021

Subject: Revised Manuscript (PONE-D-21-13113) “Therapeutic monitoring of adalimumab at non-trough levels in patients with inflammatory bowel disease”

Dear Prof. Emiko Mizoguchi:

Thank you for providing us with the opportunity to revise our manuscript (PONE-D-21-13113) as well as for the reviewers’ helpful suggestions and comments. We have addressed the concerns, comments, and questions raised by reviewers 1 and 2 and present our detailed point-by-point responses below. Revisions in the manuscript are indicated in red.

We hope that these revisions have improved the quality of our manuscript and have made it suitable for publication. We have uploaded the marked and unmarked copies of our manuscript, as requested.

Response to Reviewers

We are very grateful to the reviewers for investing their time and effort into reviewing our research. We answered each reviewers’ comment below.

Reviewer #1: Although this is a retrospective, observational study, it reveals the interesting results that non-trough ADA levels may predict the outcomes of IBD patients. The research is presented in an intelligible fashion. Several points should be clarified.

All through the study: This type of studies has been usually conducted for a separated disease, such as UC alone, or CD alone, not for combined IBD like this study. Please describe the advantages and disadvantages of this method.

Authors’ response:

We appreciate the reviewer's advice. In this study, the number of cases with UC or CD alone was small, so that we evaluated them as a combined IBD in all studies. The advantage of this method is that it increases the number of cases and improves statistical reliability. The downside is that we cannot show differences in the results between UC and CD.

We also analyzed cases with UC or CD alone. In the CD cases, the ADA continuation rate was significantly higher in the high drug level group than in the low drug level group (p = 0.003), but in the UC cases, there was no significant difference between the two groups (p = 0.190). However, the absence of significant differences with respect to the UC cases may have been due to the low case number (19) as compared to that for the CD cases (33). We have added these results to the Results section (Page 11, Line 21) and added Supplementary Figure 1 A-B.

Line 275: I would like to know the reasons why AAA did not correlate with treatment continuity in this study, which is quite different from most prior reports. Due to AAA levels? combined evaluation of UC and CD? Please describe.

Authors’ response:

We appreciate the reviewer's advice. The reasons why AAA levels were not related to treatment continuity include: 1. The sample size of our study was smaller than that of the Diamond study. 2. It is unclear whether AAA in this study is a neutralizing or a non-neutralizing antibody. Non-neutralizing antibodies may not affect the pharmacological action of anti-TNFα antibodies. In the assay of the Immundiagnostik AG (IDK monitor®) used in this study, it was difficult to distinguish neutralizing from non-neutralizing antibodies.

Reviewer #2: Kato et al. examined the correlation between anti-ADA antibody level and ADA continuity or endoscopic remission in UC and CD patients. Moreover, the authors examined the correlation between ADA level and ADA continuity or endoscopic remission. They found that a higher serum ADA level, including non-trough level, was associated with both higher ADA continuity and endoscopic remission rate, but level of AAA did not. Thus, they concluded that ADA level, including trough level and non-trough level, is useful for predicting clinical outcome of UC and CD patients treated with ADA.

Authors’ response:

We are grateful to the reviewer for evaluating this study. We have reviewed and revised the manuscript according to the reviewer’s comments as follows.

There are some concerns about this study.

1. The authors evaluated the ADA level, including trough and non-trough. In each result, they should show the timing of measuring the ADA concentration (average, mean, etc.).

Authors’ response:

We appreciate the reviewer’s precise advice. The timing of the ADA concentration measurement was 11 days after ADA administration (median, IQR 4-14). This important point in our research was added to the Results section (Page 12, Line 14).

2. In Diamond study, combination of ADA and immunomodulator more effective on mucosal improvement that ADA monotherapy. They should show the endoscopic remission rate or endoscopic improvement rate in ADA monotherapy therapy and in ADA and IM combination therapy.

Authors’ response:

We appreciate the reviewer's advice. In our study, contrary to the results of the Diamond study, there was no significant difference in the rate of mucosal healing between the ADA monotherapy group and the ADA and IM combination therapy group (p = 0.34, χ2 test). We added this result to the Discussion section (Page 16, Line 6).

3. In general, the ADA continuity is associated with the improvement of clinical symptoms. Therefore, the ADA level may be associated with the improvement of clinical symptoms. They should show the correlation between the ADA level and the improvement of clinical symptoms.

Moreover, they should show the correlation between the positive or negative of AAA and the improvement of clinical symptoms.

Authors’ response:

Unfortunately, this was a retrospective study and lacked records of clinical evaluations, such as CAI and CDAI. It was, thus, not possible to correlate the ADA concentration and positive or negative AAA with clinical symptoms. We described this issue as a limitation in the Discussion section (Page 16, Line 20). We are very sorry that we cannot amend the manuscript based on the reviewer’s good suggestions.

4. The timing of evaluation of AAA level is missing.

Authors’ response:

We appreciate the reviewer's advice. In all cases, the AAA level was evaluated at the same timing as the ADA concentration measurement. We added this information to the Methods section (Page 6, Line 19).

5. The severity of UC and CD patients in this study is missing.

Authors’ response:

We appreciate the reviewer’s suggestions. As mentioned earlier, this study lacked records for clinical evaluations such as CAI and CDAI. The endoscopic severity of 44 cases evaluated endoscopically was MES 2 for UC (median, IQR 0-3) and SES-CD 9 for CD (median, IQR 0-25). We described the endoscopic severity in the Results section (Page 12, Line 14).

We are most grateful to the reviewers' suggestions that helped to provide a more consistent and sounder interpretation of our results.

---

## [Decision Letter · Decision Letter 1]

29 Jun 2021

Therapeutic monitoring of adalimumab at non-trough levels in patients with inflammatory bowel disease

PONE-D-21-13113R1

Dear Professor Ken Sugimoto,

We’re pleased to inform you that your manuscript has been judged scientifically suitable for publication and will be formally accepted for publication once it meets all outstanding technical requirements.

Kind regards,

Emiko Mizoguchi, M.D., Ph.D.

Academic Editor

PLOS ONE

Additional Editor Comments (optional):

Reviewers' comments:

Reviewer's Responses to Questions

**Comments to the Author**

1. If the authors have adequately addressed your comments raised in a previous round of review and you feel that this manuscript is now acceptable for publication, you may indicate that here to bypass the “Comments to the Author” section, enter your conflict of interest statement in the “Confidential to Editor” section, and submit your "Accept" recommendation.

Reviewer #1: All comments have been addressed

Reviewer #2: All comments have been addressed

2. Is the manuscript technically sound, and do the data support the conclusions?

Reviewer #1: Yes

Reviewer #2: Yes

3. Has the statistical analysis been performed appropriately and rigorously? 

Reviewer #1: Yes

Reviewer #2: Yes

4. Have the authors made all data underlying the findings in their manuscript fully available?

Reviewer #1: Yes

Reviewer #2: Yes

5. Is the manuscript presented in an intelligible fashion and written in standard English?

Reviewer #1: Yes

Reviewer #2: Yes

6. Review Comments to the Author

Reviewer #1: (No Response)

Reviewer #2: (No Response)

7. PLOS authors have the option to publish the peer review history of their article (what does this mean?). If published, this will include your full peer review and any attached files.

Reviewer #1: No

Reviewer #2: No

---

## [Editor Report · Acceptance letter]

1 Jul 2021

PONE-D-21-13113R1 

Therapeutic monitoring of adalimumab at non-trough levels in patients with inflammatory bowel disease 

Dear Dr. Sugimoto:

I'm pleased to inform you that your manuscript has been deemed suitable for publication in PLOS ONE. Congratulations! Your manuscript is now with our production department. 

Kind regards, 

on behalf of

Dr. Emiko Mizoguchi 

Academic Editor

PLOS ONE